# Characterization of Key Odorants in Lushan Yunwu Tea in Response to Intercropping with Flowering Cherry

**DOI:** 10.3390/foods13081252

**Published:** 2024-04-19

**Authors:** Yinxiang Gao, Zhiyong Lei, Jigang Huang, Yongming Sun, Shuang Liu, Liping Yao, Jiaxin Liu, Wenxin Liu, Yanan Liu, Yan Chen

**Affiliations:** 1Institute of Jiangxi Oil-Tea Camellia, Jiujiang University, Jiujiang 332005, China; 4120001@jju.edu.cn (Y.G.);; 2Jiujiang Agricultural Technology Extension Center, Jiujiang 332000, China; 3Jiangxi Institute of Red Soil and Germplasm Resources, Nanchang 330046, China

**Keywords:** Lushan Yunwu tea, flowering cherry, volatile compounds, key aroma compounds, HS-SPME/GC–MS

## Abstract

Lushan Yunwu tea (LSYWT) is a famous green tea in China. However, the effects of intercropping tea with flowering cherry on the overall aroma of tea have not been well understood. In this study, headspace solid-phase microextraction (HS-SPME) coupled with gas chromatography–mass spectrometry (GC–MS) was used for analysis. A total of 54 volatile compounds from eight chemical classes were identified in tea samples from both the intercropping and pure-tea-plantation groups. Principal component analysis (PCA), orthogonal partial least-squares discriminant analysis (OPLS-DA), and odor activity value (OAV) methods combined with sensory evaluation identified cis-jasmone, nonanal, and linalool as the key aroma compounds in the intercropping group. Benzaldehyde, α-farnesene, and methyl benzene were identified as the main volatile compounds in the flowering cherry using headspace solid-phase microextraction/gas chromatography–mass spectrometry (HS-SPME/GC–MS). These findings will enrich the research on tea aroma chemistry and offer new insights into the product development and quality improvement of LSYWT.

## 1. Introduction

The tea plant (*Camellia sinensis*) is a shrub or tree distributed in tropical and subtropical regions that belongs to the genus *Camellia* in the Camellia family. The forests of Southwest China are the center of origin of tea plants [1]. Tea leaves are a popular source of tea. The growth and quality of tea leaves are affected by soil nutrition, environmental factors, cultivation methods, and other factors [2,3]. Traditional tea plantations have gradually encountered problems such as soil acidification, increased pest and disease pressure, and decreased yields [4,5,6]. The intercropping model of growing tea trees and other species can maximize land resource benefits, enhance the resistance of tea trees to adversity, and improve the biodiversity of tea gardens, thereby enhancing their comprehensive benefits for ecology, landscapes, and economies [7,8,9,10].

Intercropping models have been widely used in tea cultivation to improve the quality of tea. Recent studies found that many intercropping plants, including herbaceous plants, shrubs, and woody plants, can promote the growth of tea plants and improve their quality [1,11,12]. Chestnut–tea intercropping can reduce the environment and soil temperatures and increase the air and soil moisture in tea plantations [1]. Intercropping tea with soybean, corn, and rubber can also retain soil moisture and reduce surface erosion [13,14]. Soybean–tea intercropping can affect the content of available nitrogen in the soil and significantly increase the content of ammonium nitrogen [15]. Zhang et al. [16] found that volatile compounds released by interplanting *Cassia tora* and *Leonurus artemisia* in tea plantations can repel green tea leafhoppers. Chinese chestnut–tea intercropping was found to decrease the content of tea polyphenols and the bitter taste of tea and increase the content of amino acids and the freshness of green tea [1]. Soybean–tea intercropping was found to increase the content of non-ester-type catechins, total catechins, and theanine; decrease the content of ester-type catechins; and improve tea quality [17]. Recent studies showed that pea–tea intercropping can improve tea quality by regulating amino acid metabolism and flavonoid biosynthesis [12].

Studies showed that interplanting tea with other plants can promote the biosynthesis of aroma compounds, thereby improving the aroma of tea. Intercropping chestnut trees with tea increases the aroma of green tea by increasing phenylalanine-related metabolites and jasmine lactones [1]. Yang et al. [18] found that the intercropping of *Prunus salicina lindl.* in tea gardens improved the chestnut and floral aroma of green tea. The planting patterns of tea trees intercropped with other plants can be traced back to the Eastern Jin Dynasty in ancient China [11]. During the Song Dynasty, “Tea Garden” recorded a tea plant company with plums belonging to the Rosaceae family. In a study on companion plants in ancient, forest, and terraced tea gardens, Rosaceae was found to be one of the dominant families [11]. Flowering cherry (*Cerasus* sp.) belongs to the genus *Prunus* of the Rosaceae family. It is a famous ornamental woody plant that generally flowers in April and is cultivated worldwide [19].

Lushan Yunwu tea (LSYWT) is one of the top ten famous green teas in China, originating from Lushan Mountain, Jiangxi Province, China [20]. LSYWT has a history of more than 1800 years. As a new cultivation model, intercropping tea with flowering cherry has received little attention regarding its influence on tea aroma. In this study, 54 volatile compounds were identified using HS-SPME/GC–MS from the tea samples of the intercropping group (EG) and the pure tea tree group (CG). Through principal component analysis (PCA), orthogonal partial least-squares discriminant analysis (OPLS-DA), and odor activity value (OAV) methods, combined with sensory evaluation, we found that the EG green tea exhibited a floral aroma, while the CG did not possess a floral aroma. We also identified volatile compounds in flowering cherry using HS-SPME/GC–MS analysis and investigated the potential factors contributing to the production of key fragrance compounds, such as EG. The results of this study provided information on the aroma-active compounds in Lushan Yunwu green tea intercropped with flowering cherry. The findings of our study provide meaningful guidance for aroma quality control in Lushan Yunwu green tea.

## 2. Materials and Methods

### 2.1. Plant Materials and Growth Conditions

Fresh tea leaves from the cultivar “Population” with one leaf and one bud were collected from the Hailu Tea Expo Garden (116°3′38″ E, 29°34′114″ N) in Jiujiang, Jiangxi on 2 April 2023. The line spacing of the tea trees was 1.5 m, crowns were 70–90 cm, and plant height was 1–1.2 m. The spacing between the flowering cherry trees was 4 m, row spacing was 6 m, the crown breadth was 4–5 m, and the tree height was 4.5–5.5 m. Two sample plots were set up, one of which involved interplanting flowering cherry trees with tea trees. Tea trees within a 3 m radius of a well-growing cherry blossom tree were selected, and one bud and one leaf were randomly collected from each tea tree. The second sample comprised pure tea trees. Tea trees with good growth were selected, and one bud and leaf were randomly collected from each tea tree. LSYWT was produced using traditional processing methods, including fixing, rolling, and roasting, by an inheritor of the intangible cultural heritage of a tea master. The tea samples were stored at −40 °C until further analysis was performed.

### 2.2. Sensory Assessment

The tea sensory evaluation adopted a quantitative descriptive analysis (QDA) method [21]. The team members systematically evaluated the typical aroma types and descriptive vocabulary of green tea before performing a quantitative descriptive analysis. The evaluation panel consisted of four males and five females, aged 20–44 years. During the preparatory phase, the aroma attributes of the samples were discussed until all members agreed on their properties. Flowery, chestnut, tender, fresh, clear, and grass were selected as indicators for the quantitative description. The reference system was established and training was conducted according to the method of Wang et al. [22] until the members were familiar with the odor characteristics and intensity of these five aroma attributes [23].

Quantitative descriptive analysis was used to evaluate the intensity of various aroma characteristics with an intensity scale ranging from 1 to 9, where 1 represents the weakest aroma and 9 represents the strongest aroma. Tea samples (3 g) were added to 150 mL of boiling water and the tea infusion was obtained after 4 min. Subsequently, the tea soup was poured into a matching bowl and the cylindrical cup and matched bowl were randomly numbered for blind review. The evaluation panel repeated the process for each sample thrice, and the results were averaged. Sensory evaluation data collection and use were in accordance with a Jiujiang University Human Ethics application, ID Number: JJUM20240090.

### 2.3. Chemicals

β-Myrcene (≥90.0%) and cis-3-Hexenyl hexanoate (98%) were purchased from Macklin Biochemical Co. Ltd. (Shanghai, China). Benzylalcohol (≥99.5%), Linalool (98%), Nonanal (96%), α-Terpineol (>95.0%), Geraniol (≥99.0%), cis-Jasmone (98%), α-Farnesene (98%), and methyl palmitate (99%) were obtained from Aladdin Biochemical Co. Ltd. (Shanghai, China). Linalool oxide (pyranoid) and trans-Nerolidol were purchased from Yuanye Bio-Technology Co. Ltd. (Shanghai, China). *n*-Alkane mixtures of C7–C40 were supplied by Sigma-Aldrich Trading Co. Ltd. (Shanghai, China) and used to determine retention indices (RI). Ethyl decanoate was used as the internal standard and purchased from Aladdin Biochemical Co. Ltd. (Shanghai, China). Ultra-pure water was obtained using a Milli-Q purification system (Millipore, Bedford, MA, USA). Distilled water was purchased from the Hangzhou Wahaha Group Co. Ltd. (Hangzhou, China).

### 2.4. Extraction of Volatile Compounds from Tea Using HS-SPME

Volatile compounds were extracted using HS-SPME with a divinylbenzene/carboxen/polydimethylsiloxane coating fiber (50/30 μm; 1 cm) purchased from Supelco (Bellefonte, PA, USA) [24]. All tea samples were ground, passed through 40 mesh, and sealed for future use. Briefly, ground tea powder (1.5 g) was accurately weighed into a 20 mL headspace flask. Then, NaCl (0.5 g) was added, followed by 20 μL of ethyl decanoate (20 μg/mL) as an internal standard, and 5.25 mL of distilled water. Subsequently, the autosampler was stabilized to ensure injection stability. The flask was incubated at 70 °C for 30 min with shaking (5 s on and 2 s off). The coating fiber was then inserted into the headspace of the flask for extraction at 70 °C for 30 min. Finally, the fiber was inserted into the injector port of the gas chromatography–mass spectrometer (GC–MS) for desorption at 250 °C for 5 min, after which GC–MS separation and identification were performed [24].

### 2.5. Gas Chromatography–Mass Spectrometry Analysis

GC–MS was performed using a Thermo TSQ 8000 spectrometer equipped with a TriPlus RSH three-in-one automatic sampler TPR (Thermo Fisher Scientific, Waltham, MA, USA). A TG-5MS gas chromatography column with 30 m × 0.25 mm × 0.25 μm film thickness (Thermo Fisher, USA) was used for separation and high-purity helium (≥99.999%) was used as carrier gas. The constant flow rate was 1.0 mL/min, and the injector temperature was 280 °C. The mass spectrometry conditions were as follows: electron bombardment ion source (EI) temperature of 300 °C, mass interface temperature of 280 °C, and ionization voltage of 70 eV. The mass-scanning range was 33 to 550 *m*/*z*. The initial oven temperature was maintained at 40 °C for 2 min and then increased at 5 °C/min to 85 °C, where it was held for 2 min. Subsequently, the temperature was increased at a rate of 2 °C/min to 110 °C and held for 2 min, then increased at a rate of 4 °C/min to 220 °C, where it was held for 2 min, and finally increased at 5 °C/min to 250 °C and held for 10 min.

### 2.6. Identification and Quantitation of Volatiles

Peak identification was performed by searching the mass spectra in the National Institutes of Standards and Technology Mass Spectrometry (NIST MS) library and comparing their retention indices (RIs) with published data using the same capillary column [25]. An *n*-alkane standard (10 µg/mL) was injected under the same GC conditions as the external standard. The volatile compounds were preliminarily identified using the similarities between the RI results, mass spectra, and published data. The relative concentrations of the released volatiles were calculated using the following formula:$$Wi=\frac{A_{i}\times W_{s}}{A_{s}}$$

*Wi* is the content of the target volatile concentration to be measured (μg/L), *A_i_* is the peak area of the target volatile concentration to be measured, *W_s_* is the concentration of the internal standard (μg/L), and *A_s_* is the peak area of the internal standard in the sample.

### 2.7. OAV Calculation

OAVs were used to assess the contributions of volatile compounds to the aroma of tea samples [26]. OAVs were calculated using the following equation:$${OAV}_{i}=\frac{C_{i}}{{OT}_{i}}$$

*C_i_* is the absolute content of volatile compounds and *OT_i_* is the odor threshold of volatile compounds in water.

### 2.8. Statistical Analysis

PCA and OPLS-DA were performed using SIMCA14.1 (Umetrics, Umea, Sweden). Significant differences between samples were assessed using one-way analysis of variance (ANOVA) with SPSS (version 20.0; SPSS Inc., Chicago, IL, USA). TBtools (version 2.056, accessed on 20 January 2024, https://github.com/CJ-Chen/TBtools) was used for heat mapping and hierarchical analysis. Data significance analysis was performed using the Student’s *t*-test and significant differences (*p* < 0.05) are denoted by *.

## 3. Results and Discussion

### 3.1. Aroma Characteristics of Teas in the Intercropping Group (EG) and Pure Tea Tree Group (CG)

Aroma is an important indicator for evaluating tea quality [27]. In this study, we found differences between the aroma profiles of EG and CG green tea using sensory descriptive analysis. As shown in the radar plot in Figure 1, the “flowery” aroma of EG green tea was significantly enhanced (*p* < 0.01), whereas the “fresh and clear” aroma was significantly weakened (*p* < 0.01); the “grass”, “chestnut”, and “tender” aromas were not significantly different (*p* > 0.05).

### 3.2. Identification and Quantification of Volatile Compounds in EG and CG Green Tea

To further explore why the floral aromas of the EG tea samples were more prominent than those of the CG samples, HS-SPME and GC–MS were used to analyze the volatile compounds in the EG and CG green teas grown under the two different cultivation modes, and 54 volatile compounds were identified (Table 1). Among them, 46 and 29 volatiles were identified in EG and CG green tea, respectively. Based on their chemical structures, these volatile compounds were divided into eight different chemical classes: 16 alcohols, 7 esters, 6 ketones, 2 aldehydes, 11 olefins, 2 aromatics, 1 heterocycle, and 9 hydrocarbons. To further analyze the volatile components of EG and CG green tea, we analyzed the percentage contents of 54 volatile components in EG and CG green tea (Figure 2). VIP values of the 54 volatile components are shown in Appendix A. The primary volatile components in EG and CG green tea were alcohols and esters, which accounted for 32.41% and 21.90% of the total volatile content in EG green tea, and 30.70% and 16.97% in CG green tea, respectively. Additionally, the hydrocarbon content in EG green tea (13.42%) was lower than that in CG green tea (17.77%). The ketone, aldehyde, and aromatic contents in EG green tea (16.14%, 6.15%, 5.08%) were higher than those in CG green tea (7.69%, 0.00%, and 0.00%, respectively). The main volatile compounds in EG green tea were cis-3-hexenylhexanoate (60.12 μg/L), cis-jasmone (46.04 μg/L), methyl salicylate (39.71 μg/L), linalool oxide (pyranoid) (36.42 μg/L), linalool (34.58 μg/L), and nonanal (30.45 μg/L). The main volatile compounds in CG green tea were linalool (76.47 μg/L), cis-3-hexenylhexanoate (67.93 μg/L), trans-nerolidol (48.91 μg/L), pentadecane (41.71 μg/L), hexadecane (34.84 μg/L), and tetradecane (32.81 μg/L).

The aroma is an important factor in evaluating the quality of tea, and the formation of the overall aroma of tea is crucial to the type and content level of volatile compounds. In this study, there were 46 types of volatile compounds in EG green tea and 29 types in CG green tea. The difference in volatile components between the EG and CG green teas may be one of the reasons why EG green tea presents a floral scent, whereas CG green tea does not. Among these volatile compounds, cis-jasmone, methyl salicylate, and nonanal had the highest content levels in EG green tea. As reported in the existing literature, these compounds are components of floral and fruit odors. The differences in the proportions and content levels of volatile compounds may be one of the reasons why the experimental group’s tea was floral in odor, whereas the control group was not. Xu et al. [28] found significant differences in the varieties and proportions of volatile compounds (alcohols, aldehydes, esters, terpenes, alkenes, and ketones) in different cultivation environments, which is consistent with the results of this experiment.

### 3.3. Key Volatile Component Analysis in EG

Figure 3 shows a heat map of the 54 volatile compounds, and PCA was performed based on the quantitative results of the aroma components. Principal components PC1 and PC2 individually accounted for 76.3% and 10.9% of the total variance (Figure 4). PCA showed a clear separation between the different groups, and the results were consistent with those of heat map cluster analysis. To further reveal the influences of cherry blossoms on key aroma compounds in tea, OPLS-DA was performed on EG and CG green tea in this study (Figure 5). The fitting index of the independent variable was R^2^X = 0.862, that of the dependent variable was R^2^Y = 0.99, and that of the prediction was Q^2^ = 0.969, indicating that the model had a good cumulative explanatory degree, predictive ability, and stability. Nine key volatile compounds were screened using the OPLS-DA model (VIP > 1, *p* < 0.05): cis-jasmone, linalool, indole, geraniol, ethylene glycol monoisobutyl ether, tetradecane, nonanal, 2-nonen-1-ol, and hexadecane.

Cis-jasmone is one of the characteristic aromas of green tea [29] and one of the main components of the plant’s direct and indirect stress-resistance mechanisms [30]. Indole in tea is produced from tryptophan, which presents a floral aroma at low concentrations and enhances the overall aroma of green tea within a certain concentration range, thus contributing to the chestnut aroma of green tea [31,32]. Geraniol and linalool are released from geranyl pyrophosphate precursors by geraniol synthase and linalool synthase, respectively, and linalool is further oxidized to form dehydrolinalool and various other linalool oxides [33]. Linalool has floral and citrus fruit aromas, whereas geraniol has a rose aroma [33]. Liu et al. [31] showed that geraniol is an important source of floral aroma. Han et al. [34] found that geraniol, linalool, cis-jasmone, and indol are the key aroma compounds in green tea, which is in agreement with the findings of this study. Nonanal is the key aroma compound in green tea and it is positively correlated with the floral aroma intensity of tea [35,36].

### 3.4. Key Aroma Compounds Identified by OAV in the EG Group

The OAV for each compound was calculated in order to assess the contribution of the volatiles to the overall aroma. Based on these nine key volatile compounds, three key aroma compounds, cis-jasmone, nonanal, and linalool, were identified (OVA > 1) (Table 2). Among these compounds, two showed OAVs > 10, indicating their substantial contributions to the overall aroma profile of EG green tea. Cis-jasmone had the highest OAV (177.07), followed by nonanal (OAV = 30.45). Cis-jasmone and nonanal were aroma compounds unique to EG. Cis-jasmone and nonanal may have contributed to the floral appearance of the EG.
foods-13-01252-t002_Table 2Table 2Aroma compounds with OAVs identified in EG and CG green tea samples.OdorantsOdor DescriptionVIPOT (μg/L)OAVcis-Jasmone *Flowery3.190.26177.07Linalool *Floral, lavender2.8862.17Tetradecane *Alkane2.2910,0000.0032Nonanal *Sweet Orange aroma2.12130.452-Nonen-1-ol *-1.98--Indole *Floral, animal-like1.76400.63Ethylene glycol monoisobutyl ether *-1.72--Hexadecane *Alkane1.0813,000,0000Geraniol *Rose1.05400.37OT—odor threshold in water based on the literature [37,38]; OAV—odor activity value; VIP—variable importance in projection. Data significance analysis was performed using Student’s *t*-test, and a significant difference (*p* < 0.05) is denoted by *.


Nine key aromatic compounds were identified using the OPLS-DA model (VIP > 1, *p* < 0.05), and the contents of these nine compounds were significantly different between the EG and CG teas. Recent studies showed that high concentrations of compounds do not necessarily produce a strong aroma. Tea typically contains hundreds of volatile compounds; however, not all volatile components play a significant role in its aroma quality [26,30,39,40]. Wang et al. [26] showed that only a few compounds determined the overall aroma quality of tea, and when some key compounds were missing, the overall aroma quality of tea would be significantly affected. The aroma of tea depends not only on the type and content of volatile compounds but also on their OAVs [41]. We used OAVs to evaluate the contribution of volatile compounds to the EG green tea and selected three key aromatic compounds (VIP > 1, *p* < 0.05, OAV > 1): cis-jasmone, linalool, and nonanal. Linalool has a citrus-like and flowery aroma [27,41]. Su et al. [42] showed that linalool is the key volatile compound of green tea and that it plays an important role in the aroma of green tea. Nonanal is widely found in green tea and has a sweet, fruity flavor [37]. Notably, cis-jasmone and nonanal were the key volatile compounds responsible for the unique floral flavor of EG green tea.

Cis-jasmone and nonanal were the main components of EG green tea. The aroma quality of tea is determined not only by its content of compounds but also by the activity value of its aroma. The unique floral-aroma compound cis-jasmone in EG green tea had not only a high content but also a high OAV of 177.07. This indicates that cis-jasmone is a characteristic aroma compound of EG green tea and contributes significantly to its overall aroma. cis-jasmone is a compound with a jasmine scent that is commonly found in green tea and produces a pleasant floral jasmine scent at lower concentrations. Zhu et al. [43] found that cis-jasmone is a characteristic aroma compound of Xihu Longjing green tea, which is consistent with the findings of this study. Katsuno et al. [32] demonstrated that cis-jasmone contributes to the formation of floral fragrances. Cis-jasmone, linalool, and nonanal, the three key aromatic compounds in EG, promote the formation of floral fragrances [44].

Cis-jasmone and nonanal are the unique aroma-characteristic compounds of EG green tea. To explain this phenomenon, we studied the growth environment of the EG. Tea plants are exposed to various natural stresses, such as light, temperature, and pests. When facing stress, tea plants release volatile organic compounds to reduce the adverse effects of the stress [45]. The EG tea trees were intercropped with flowering cherry trees. When cherry trees bloom, flowers compete for scarce resources by continuously releasing volatile compounds. During resource competition, tea trees secrete large amounts of volatile compounds to defend against stress.

### 3.5. Analysis of Volatile Compounds in Flowering Cherry Trees

The results showed that EG green tea produced floral volatile compounds, such as cis-jasmone and nonanal, and that the content of other key floral volatile compounds changed. To explain these phenomena, HS-SPME/GC–MS was used to analyze the volatile compounds present in flowering cherry flowers. The detailed results are presented in Table 3. The results showed that benzaldehyde, α-farnesene, methyl benzoate, 3, 5-dimethoxytoluene had high relative contents, and that benzaldehyde accounted for 86.12% of all aroma components.

Allelopathy was first proposed by Molisch in 1937. During growth and development, plants change their surrounding microecological environment by releasing specific metabolites that affect the growth and development of other plants [46]. Allelopathy includes promotion and inhibition [47]. Allelopathic compounds can inhibit seed germination and slow or destroy normal plant growth. Therefore, the use of allelopathy to remove weeds is an effective method [48]. Studies showed that many allelopathic compounds can stimulate plant growth. The leaf extracts of *Pinus sylvestris* and *Broussonetia papyrifera* have been shown to promote the growth of *Amygdalus pedunculata* seedlings [49]. Research has shown that leaf extracts from *Hippophae rhamnoides*, *Hedysarum mongolicum*, and *Sabina vulgaris* significantly promote the seed germination and seedling growth of *Amygdalus pedunculata* Pall [50]. Plant allelochemicals are mainly secondary metabolites distributed in the roots, stems, leaves, flowers, fruits, and seeds of plants. Plant secondary metabolites include phenols, alkaloids, organic acids, terpenoids, and glycosides.

We analyzed the volatile compounds present in the flowers of the flowering cherry. Benzaldehyde had the highest concentration, followed by α-farnesene. Benzaldehyde is the simplest aromatic aldehyde found in nature, and it is present not only in >50% of plant families but also in insects and non-insect arthropods [51]. Benzaldehyde acts as a pollinator attractant, fragrance volatiler, and an antifungal compound in plants [52]. Benzaldehyde was found to be one of the main volatiles in sweet cherry flowers, which is consistent with the results of Zhang et al. [53]. Recent studies found that in the interplanting system between oil tea and peanuts, oil tea releases benzaldehyde as an allelopathic compound that inhibits the germination of peanut seeds [54]. α-farnesene has a citrusy odor and is involved in signal transduction between tea plants; it promotes metabolite synthesis in adjacent undamaged tea leaves [55]. Studies showed that α-farnesene acts as a signal to activate the expression of β-1, 3-glucanase (CsBGL) gene in adjacent undamaged tea leaves [56]. Methyl benzoate has a fruity odor and is a common floral volatile component found in more than 80 plants [57,58]. We speculated that benzaldehyde, α-farnesene, and methyl benzoate in flowering cherry may have allelopathic interactions with tea trees as major allelopathic compounds, which affect the metabolism of tea trees and prompt EG green tea trees to produce key volatile compounds with unique flower flavors (Figure 6). The results showed that a system of tea tree intercropping with flowering cherry has the potential to improve the tea quality.

There were significant differences between the volatile compounds of tea under the two cultivation models, indicating that the intercropping model had a significant influence on the secondary metabolism of tea. These results enhance our understanding of the improvement in tea quality created by the model with tea intercropping with the flowering cherry, and they provide a reference for the future popularization and application of tea intercropping in the flowering cherry model.

## 4. Conclusions

In this study, HS-SPME/GC–MS was used to identify volatile compounds of EG and CG green tea, and 54 volatile compounds were identified. The 54 volatile compounds were identified using PCA, OPLS-DA, and OAV. Among these compounds, cis-jasmone, linalool, and nonanal were found to be the key aromatic compounds responsible for EG green tea’s floral scent production. It was speculated that benzaldehyde, α-farnesene, and methyl benzene in flowering cherry trees acted as major allelopathic compounds, affecting the metabolism of tea trees through allelopathy. Cis-jasmone, linalool, and nonanal promoted the formation of a floral aroma and improved the overall aroma quality of tea. Further exploration is needed to determine whether other flowering plants affect the aroma quality of tea.

## Figures and Tables

**Figure 1 foods-13-01252-f001:**
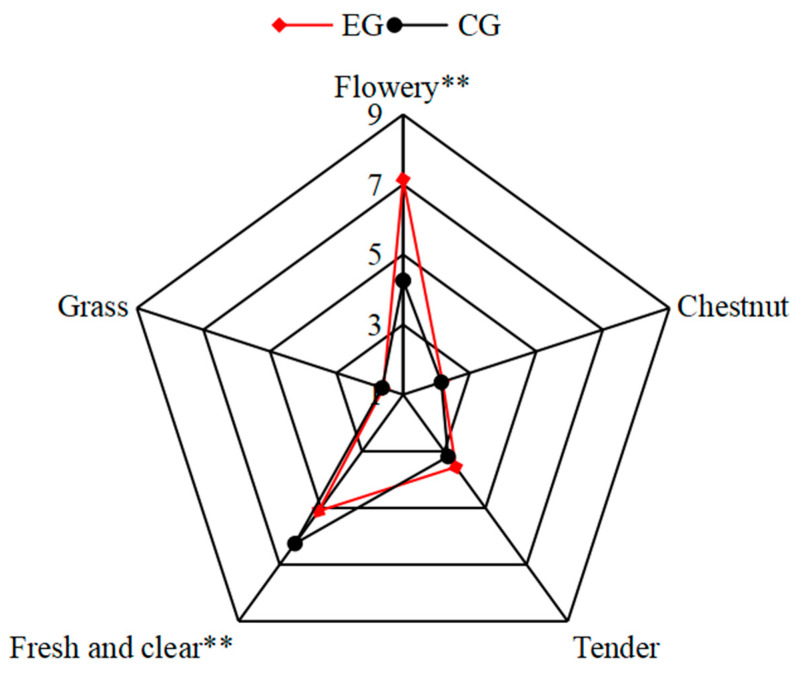
Radar plot for comparative descriptive aroma profiles of the EG and CG green teas. (EG—teas from the intercropping group; CG—teas from the pure tea tree group. ** indicates *p* < 0.01).

**Figure 2 foods-13-01252-f002:**
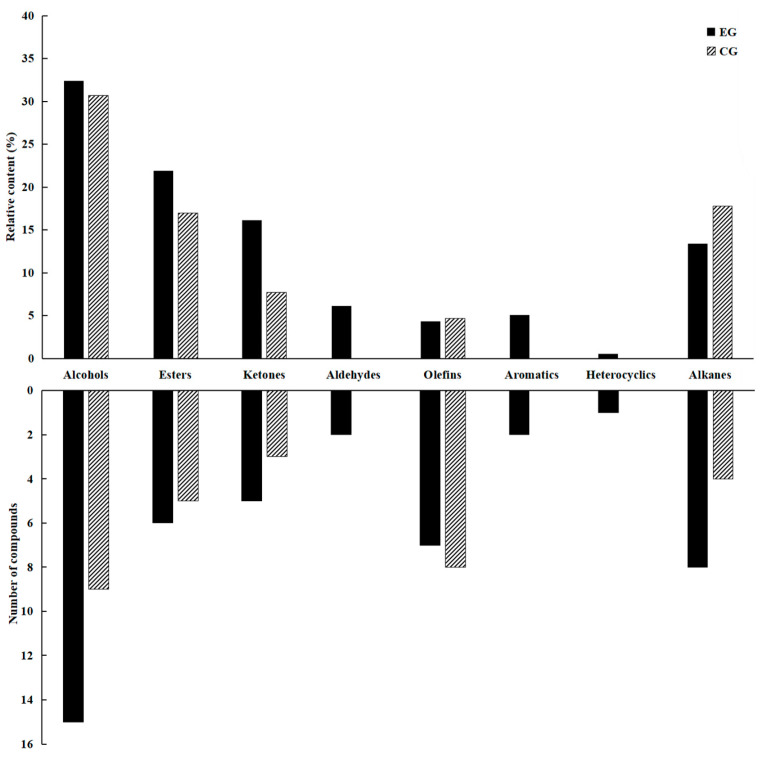
Volatile profiles of EG and CG green tea determined by SPME-GC–MS. The upper part of the figure shows the relative content of each chemical category, and the lower part shows the number of different compounds in each category.

**Figure 3 foods-13-01252-f003:**
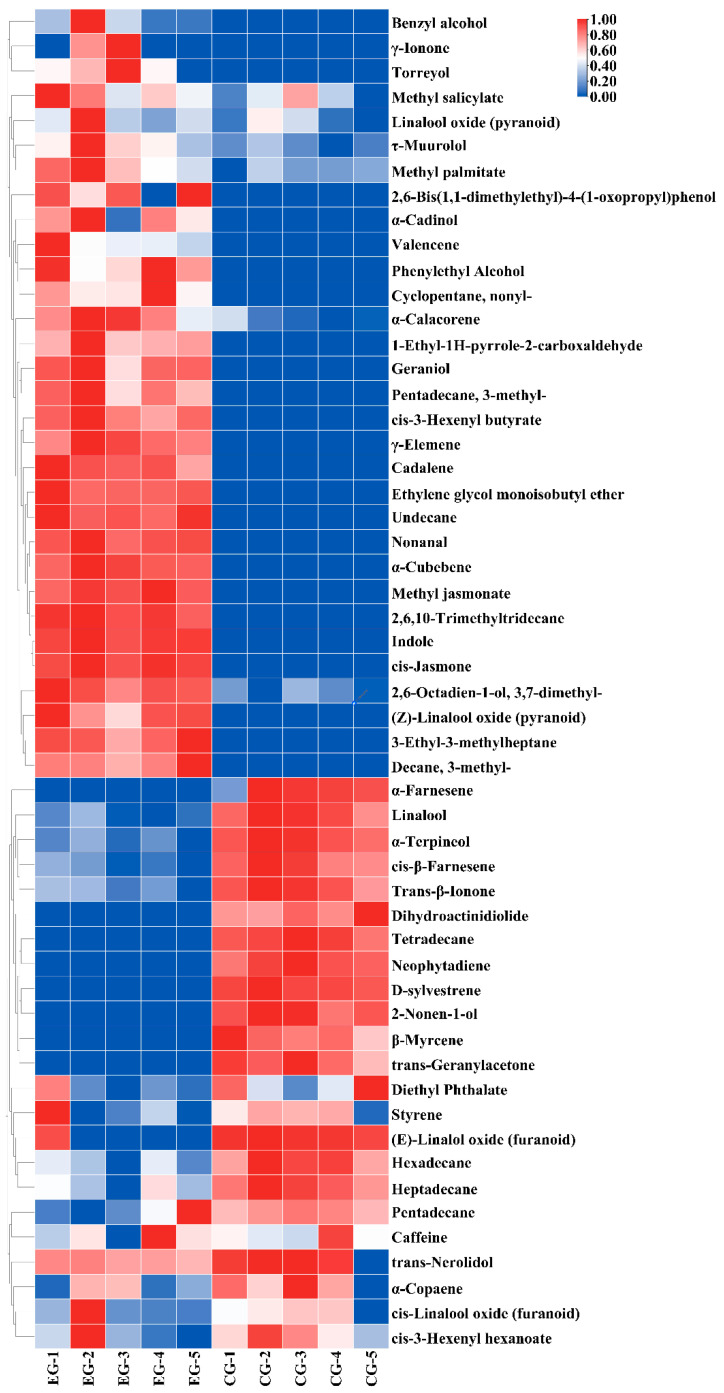
Heat map analysis of the concentrations of the main volatile compounds in tea samples.

**Figure 4 foods-13-01252-f004:**
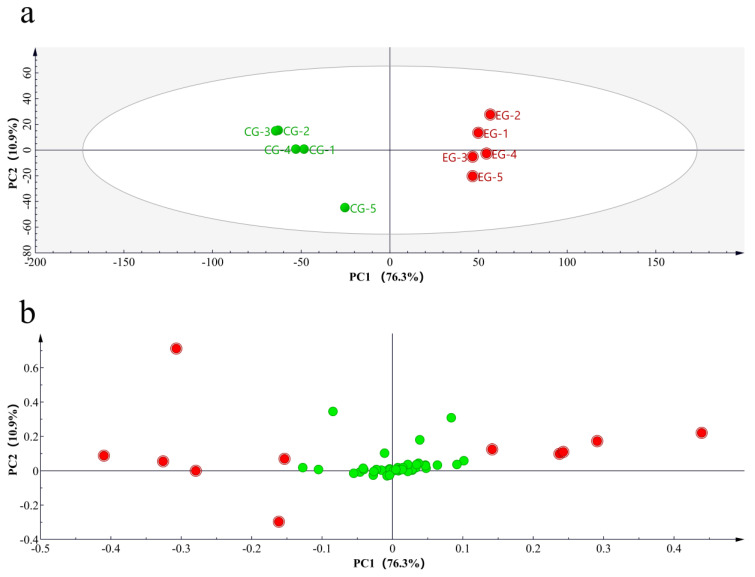
Multivariate analysis of volatile compound contents between the EG and CG green tea samples. (**a**) PCA score plot and (**b**) loading plots of volatile compounds (red and green dots represent different compounds; red dot represent the most differential compounds, VIP > 1).

**Figure 5 foods-13-01252-f005:**
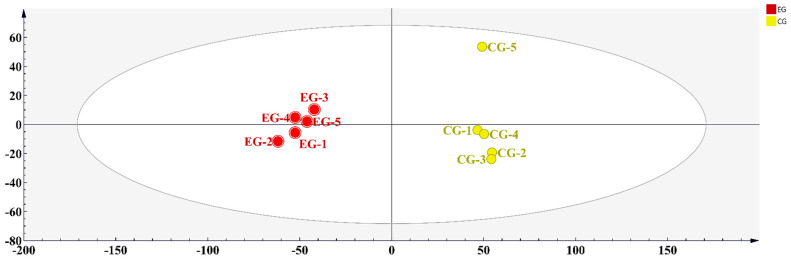
The OPLS-DA model in the EG and CG green tea samples (R^2^X = 0.862, R^2^Y = 0.99, and Q^2^ = 0.969).

**Figure 6 foods-13-01252-f006:**
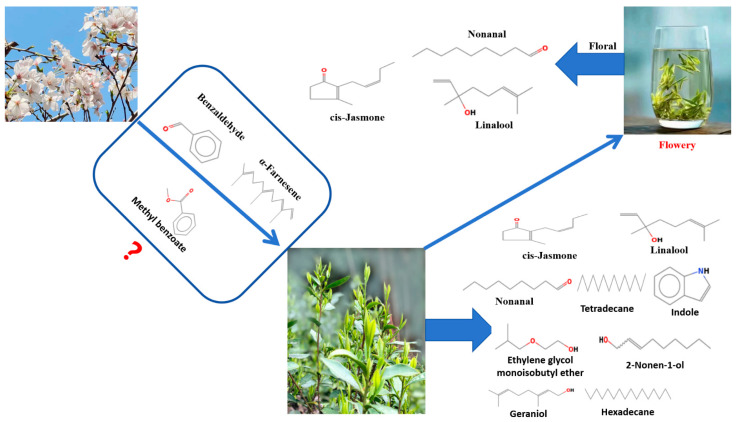
Model of Lushan Yunwu tea intercropping with flowering cherry leading to floral tea. We speculated that benzaldehyde, α-farnesene, and methyl benzoate in flowering cherry may have allelopathic interactions with tea trees as major allelopathic compounds, which affect the metabolism of tea trees and prompt leaves to produce key volatile compounds with unique flower flavors.

**Table 1 foods-13-01252-t001:** Volatile compounds identified in EG and CG green tea (µg/L).

	RT	Name	CAS	RI ^a^	RI ^b^	ID	EG	CG
1	7.57	Styrene	100-42-5	897	893	MS, RI	2.52 ± 0.6	2.87 ± 0.38
2	8.8	Ethylene glycol monoisobutyl ether	4439-24-1	940	947	MS, RI	25.22 ± 5.49	-
3	9.03	3-Ethyl-3-methylheptane	17302-01-1	959	953	MS, RI	3.09 ± 0.59	-
4	10.34	β-Myrcene	123-35-3	992	991	MS, RI, STD	-	0.85 ± 0.16
5	11.67	D-sylvestrene	1461-27-4	1030	1027	MS, RI	-	4.85 ± 0.37
6	12.14	Benzyl alcohol	100-51-6	1040	1036	MS, RI, STD	0.77 ± 0.89	-
7	12.49	1-Ethyl-1H-pyrrole-2-carboxaldehyde	2167-14-8	1048	1046	MS, RI	2.27 ± 0.74	-
8	12.85	Decane, 3-methyl-	13151-34-3	1061	1071	MS, RI	2.98 ± 0.69	-
9	13.3	cis-Linalool oxide (furanoid)	5989-33-3	1073	1074	MS, RI	12.99 ± 3.14	13.9 ± 1.89
10	13.95	(E)-Linalol oxide (furanoid)	34995-77-2	1090	1886	MS, RI	2.72 ± 5.44	15.91 ± 0.96
11	14.16	Undecane	1120-21-4	1095	1100	MS, RI	2.83 ± 0.31	-
12	14.38	Linalool	78-70-6	1101	1098	MS, RI, STD	34.58 ± 3.73	76.47 ± 6.14
13	14.58	Nonanal	124-19-6	1105	1102	MS, RI, STD	30.45 ± 5.61	-
14	14.61	2-Nonen-1-ol	22104-79-6	1105	1105	MS, RI	-	28.57 ± 6.49
15	15.36	Phenylethyl Alcohol	60-12-8	1122	1116	MS, RI	0.88 ± 0.31	-
16	17.69	(Z)-Linalool oxide (pyranoid)	14009-71-3	1170	1174	MS, RI	5.04 ± 1.64	-
17	17.85	Linalool oxide (pyranoid)	14049-11-7	1175	1178	MS, RI, STD	36.42 ± 5.53	31.75 ± 3.51
18	18.43	cis-3-Hexenyl butyrate	16491-36-4	1187	1187	MS, RI	4.94 ± 1.15	-
19	18.72	α-Terpineol	98-55-5	1193	1190	MS, RI, STD	3.29 ± 0.6	14.12 ± 1.32
20	19.05	Methyl salicylate	119-36-8	1200	1192	MS, RI	39.71 ± 7.82	29.55 ± 6.78
21	22.03	Geraniol	106-24-1	1255	1255	MS, RI, STD	14.64 ± 5.51	-
22	22.24	2,6-Octadien-1-ol, 3,7-dimethyl-	624-15-7	1259	1260	MS, RI	5.68 ± 0.8	0.66 ± 0.32
23	24.45	Indole	120-72-9	1300	1295	MS, RI	25.52 ± 2.54	-
24	26.67	γ-Ionone	79-76-5	1338	1340	MS, RI	0.09 ± 0.12	-
25	27.36	α-Cubebene	17699-14-8	1350	1352	MS, RI	6.77 ± 0.89	-
26	28.87	α-Copaene	3856-25-5	1376	1376	MS, RI	1.89 ± 0.25	2.17 ± 0.33
27	29.17	cis-3-Hexenyl hexanoate	31501-11-8	1382	1380	MS, RI, STD	60.12 ± 10.38	67.93 ± 6.69
28	30.21	Tetradecane	629-59-4	1400	1400	MS, RI	-	32.81 ± 6.97
29	30.24	cis-Jasmone	488-10-8	1400	1394	MS, RI, STD	46.04 ± 6.37	-
30	32.18	γ-Elemene	29873-99-2	1434	1434	MS, RI	3.5 ± 0.64	-
31	32.93	Cyclopentane, nonyl-	2882-98-6	1447	1451	MS, RI	2.89 ± 1.53	-
32	33.32	trans-Geranylacetone	3796-70-1	1454	1453	MS, RI	-	1.63 ± 0.32
33	33.54	cis-β-Farnesene	28973-97-9	1458	1457	MS, RI	3.13 ± 0.38	7.29 ± 0.66
34	33.77	2,6,10-Trimethyltridecane	3891-99-4	1462	1461	MS, RI	10.61 ± 1.26	-
35	34.87	Valencene	4630-07-3	1481	1476	MS, RI	3.92 ± 3.74	-
36	35.24	Trans-β-Ionone	79-77-6	1487	1486	MS, RI	4.18 ± 0.48	8.41 ± 0.66
37	35.97	Pentadecane	629-62-9	1500	1500	MS, RI	26.28 ± 19.46	41.71 ± 4.58
38	36.52	α-Farnesene	502-61-4	1510	1508	MS, RI, STD	-	5.65 ± 2.59
39	37.67	Dihydroactinidiolide	17092-92-1	1530	1532	MS, RI	-	3.02 ± 0.7
40	38.47	α-Calacorene	21391-99-1	1544	1542	MS, RI	1.42 ± 0.34	0.43 ± 0.17
41	39.63	trans-Nerolidol	40716-66-3	1565	1564	MS, RI, STD	22.81 ± 4.33	48.91 ± 24.79
42	39.97	Pentadecane, 3-methyl-	2882-96-4	1570	1570	MS, RI	3.59 ± 1.27	-
43	41.48	Diethyl Phthalate	84-66-2	1598	1603	MS, RI	0.89 ± 0.97	2.02 ± 1.35
44	41.63	Hexadecane	544-76-3	1600	1600	MS, RI	19.33 ± 3.3	34.84 ± 4.15
45	43.75	2,6-Bis(1,1-dimethylethyl)-4-(1-oxopropyl)phenol	14035-34-8	1639	1640	MS, RI	2.41 ± 1.45	-
46	43.99	τ-Muurolol	19912-62-0	1643	1642	MS, RI	5.63 ± 1.42	3.39 ± 0.42
47	44.24	Torreyol	19435-97-3	1648	1645	MS, RI	0.16 ± 0.1	-
48	44.39	Methyl jasmonate	1211-29-6	1650	1652	MS, RI	9.58 ± 1.34	-
49	44.68	α-Cadinol	481-34-5	1656	1653	MS, RI	1.49 ± 0.87	-
50	45.79	Cadalene	483-78-3	1676	1674	MS, RI	1.48 ± 0.23	-
51	47.11	Heptadecane	629-78-7	1700	1700	MS, RI	2.63 ± 0.65	4.96 ± 0.49
52	54.33	Neophytadiene	504-96-1	1838	1837	MS, RI	-	2.76 ± 0.32
53	54.67	Caffeine	50-8-2	1847	1840	MS, RI	33.13 ± 9.42	2.76 ± 0.34
54	58.75	Methyl palmitate	112-39-0	1928	1926	MS, RI, STD	1.21 ± 0.28	0.68 ± 0.11

MS—identification based on the NIST 2017 mass spectral database; ^a^ RI—retention index of each volatile on the TG-5MS column (30 m × 0.25 mm × 0.25 μm); ^b^ RI—retention index based on the NIST database. STD—means the compound was identified by standards. All data are expressed as mean ± S.D. (*n* = 5).

**Table 3 foods-13-01252-t003:** The volatile compounds present in flowering cherry flowers.

Name	CAS	Relative Content (%)
Benzaldehyde	100-52-7	86.12
α-Farnesene	502-61-4	4.49
Methyl benzoate	93-58-3	3.15
Orcinol dimethyl ether	4179-19-5	2.01
Anisic aldehyde	123-11-5	0.81
Phenylethyl Alcohol	60-12-8	0.75
3,5-Dimethoxybenzaldehyde	7311-34-4	0.53
Benzene, 1-methoxy-2-(methoxymethyl)-	21998-86-7	0.1
Benzoic acid, ethyl ester	93-89-0	0.07
Methyl salicylate	119-36-8	0.07

## Data Availability

The original contributions presented in the study are included in the article/Appendix A, further inquiries can be directed to the corresponding authors.

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
