# Peer review of "Characterization of Key Odorants in Lushan Yunwu Tea in Response to Intercropping with Flowering Cherry"

_foods, 2024, doi:10.3390/foods13081252_

Round 1

Reviewer 1 Report

Comments and Suggestions for Authors

The paper deals with the effect of inter-cropping tea with flowering cherry on the overall aroma of tea.

In order to study this effect, the authors characterize the samples by means of sensory analysis and volatile compounds.

I suggest that this paper be re-written with additional information mainly regarding the sensory analysis results.  The paper needs revision and more detailed information about the design of sensory analysis performed and the statistical analysis of the data is needed.

Some important points to be reviewed are as follows:

  1. Section dealing with sensory analysis is only descriptive and in the material and methods important information is lacking (no design, no order of presentation of the samples, why the judges don’t quantify the descriptors included in the profile, how the samples were presented to the assessors, how many sessions carried out,….)
  2. The authors apply PCA but don’t include in it the sensory evaluation. My suggestion is if is possible to include a quantification of the descriptors and include this in the plot. I this way will be possible to relate the main sensory characteristics of each sample with their sensory attributes.
  3. Regarding the data analysis, the authors states that there are differences, but no level of p (level of significance) is included in the material and methods. Please, specify or clarify this point.
  4. It is necessary to review all text in order to correct spaces and ‘.’ in the manuscript, also uppercase and lowercase letters.
  5. A suggestion regarding ‘floral scent’: floral nuances?
  6. In L245 the authors refer ‘aroma quality’. Which is the meaning?

Summary

The paper needs to be improved (MAJOR revision) because I detected important information lacking in the principles of the work that is a barrier to reaching the aim of a scientific paper and drawing conclusions on it mainly related with the sensory analysis evaluation.

Author Response

Manuscript ID: foods-2958910

Dear Reviewer,

Thanks a lot for your good suggestion. We find that the comments are of great help for the improvement of our manuscript. All of the comments have been considered in the revised manuscript, and all of the revisions are marked in yellow in the revised version. Below we have provided in detail the point-by-point replies to the comments.

Best wishes,

Yours sincerely,

Yanan Liu

Point 1: Section dealing with sensory analysis is only descriptive and in the material and methods important information is lacking (no design, no order of presentation of the samples, why the judges don’t quantify the descriptors included in the profile, how the samples were presented to the assessors, how many sessions carried out,….)

Response: Thank you for your suggestion. We re-conducted the sensory evaluation between EG and CG tea samples. We added the description in the new revision (lines 97 to 114, lines 180 to 187).

Point 2: The authors apply PCA but don’t include in it the sensory evaluation. My suggestion is if is possible to include a quantification of the descriptors and include this in the plot. In this way will be possible to relate the main sensory characteristics of each sample with their sensory attributes.

Response: Thanks a lot for your good suggestion! Radar plot for comparative descriptive aroma profiles of EG and CG were added in the new revision (Figure 1, lines 182 to 187). We modified Figure 4 with loading plot (lines 261 to 263).

Point 3: Regarding the data analysis, the authors states that there are differences, but no level of p (level of significance) is included in the material and methods. Please, specify or clarify this point.

Response: Thanks a lot! We added the description in the new revision (lines 176 to 178).

Point 4: It is necessary to review all text in order to correct spaces and ‘.’ in the manuscript, also uppercase and lowercase letters.

Response: Thank you very much for putting forward this good comment! We have corrected these problems.

Point 5: A suggestion regarding ‘floral scent’: floral nuances?

Response: Thanks a lot! We added the description in the new revision (lines 182 to 187).

Point 6: In L245 the authors refer ‘aroma quality’. Which is the meaning?

Response: Thank you very much for putting forward this good comment! I’m sorry for the confused expression. We have added the description in the new revision (lines 191 to 192).

Reviewer 2 Report

Comments and Suggestions for Authors

The article ‘The article ‘Characterization of key odorants in Lushan Yunwu tea in response to intercropping with flowering cherry, presented for review is interesting. The possibility of improving the quality of teas offered to consumers is an interesting issue. The nen effect can be achieved, among other things, by modifying fermentation or extraction methods. The possibility of improving organoleptic values at the cultivation stage is an interesting approach that has been of interest to scientists for several years. However, some issues require clarification. Below are some comments that may improve the article.

- The title of the article reflects the content well.

- The summary is correct and presents the content of the work well.

- The information contained in the introduction provides a good introduction to the subject matter.

- In the reviewer's opinion, the aim of the work at the end of the introduction should be reworded. In its current form, the authors limited the purpose of the work to one sentence, while excessively developing the research results. The purpose of the work should clearly specify why the research was undertaken. The achieved results are included in the summary, discussion and summary.

- How many repetitions were individual analyzes performed?

- The sensory analysis was performed by five people, were all assessments identical? Did any of the experts pay special attention to any of the features?

- Discussion of the results and discussion of the obtained results with the literature is correct.

- Correct inference, refers to the results achieved.

Author Response

Dear Reviewer,

Thanks a lot for your good suggestion. We find that the comments are of great help for the improvement of our manuscript. All of the comments have been considered in the revised manuscript, and all of the revisions are marked in yellow in the revised version. Below we have provided in detail the point-by-point replies to the comments.

Best wishes,

Yours sincerely,

Yanan Liu

Point 1: In the reviewer's opinion, the aim of the work at the end of the introduction should be reworded. In its current form, the authors limited the purpose of the work to one sentence, while excessively developing the research results. The purpose of the work should clearly specify why the research was undertaken. The achieved results are included in the summary, discussion and summary.

Response: Thank you for your suggestion. We added the description in the new revision (lines 78 to 81).

Point 2: How many repetitions were individual analyzes performed?

Response: Thanks a lot for your good suggestion! Five repetitions were performed. We added the description in the new revision (lines 213).

Point 3: The sensory analysis was performed by five people, were all assessments identical? Did any of the experts pay special attention to any of the features?

Response: Thanks a lot! We re-conducted the sensory evaluation between EG and CG tea samples, and the panel consisted of 9 people. Radar plot for comparative descriptive aroma profiles of EG and CG were added in the new revision (lines 97 to 114, lines 180 to 187).

Reviewer 3 Report

Comments and Suggestions for Authors

This manuscript addresses key flavour compounds in teas using HS-GC-MS and statistical analysis.  The authors compared difference between intercropping and pre tea tree group using sensory analysis, GC analysis, PCA and OPLS-DA.  The authors showed that nine compounds among 54 compounds contribute to the quality.  This manuscript provides insights of effect of intercropping such as “plant talk”, however, some minor revisions should be needed, in particular Tables and Figures because of lack of abbreviation and explanation.

L193: Please add unit of EG and CG in Table 2.  Also, please explain both RIa and RIb.  Perhaps, RIa and RIb mean retention index from some database and measured value, respectively.

L221: Readers can understand that the authors selected nine compounds with variable importance of projection.  But this data could be most important part of this manuscript, so please show VIPs of all compounds in new table.

L227: Please modify Figure 3 with loading plot or biplot.  This might provide some candidates and trends in intercropping tea.

L239: Please modify Table 3.  The authors should demonstrate references of threshold and explanation of abbreviations including VIP, OT and OAV.  Also, please change p-value to p.

Author Response

Dear Reviewer,

Thanks a lot for your good suggestion. We find that the comments are of great help for the improvement of our manuscript. All of the comments have been considered in the revised manuscript, and all of the revisions are marked in yellow in the revised version. Below we have provided in detail the point-by-point replies to the comments.

Best wishes,

Yours sincerely,

Yanan Liu

Point 1: L193: Please add unit of EG and CG in Table 2. Also, please explain both RIa and RIb. Perhaps, RIa and RIb mean retention index from some database and measured value, respectively.

Response: Thank you for your suggestion. We added unit of EG and CG in Table 2. RIa mean retention index of each volatile on the TG-5MS column. RIb mean retention index based on the NIST database. We added the description in the new revision (lines 211 to 213).

Point 2: L221: Readers can understand that the authors selected nine compounds with variable importance of projection. But this data could be most important part of this manuscript, so please show VIPs of all compounds in new table.

Response: Thanks a lot for your good suggestion! We added VIPs of all compounds in new figure (Figure S1).

Point 3: L227: Please modify Figure 3 with loading plot or biplot. This might provide some candidates and trends in intercropping tea.

Response: Thank you very much for putting forward this good comment! We modified Figure 4 with loading plot (lines 261 to 263).

Point 4: L239: Please modify Table 3. The authors should demonstrate references of threshold and explanation of abbreviations including VIP, OT and OAV. Also, please change p-value to p.

Response: Thank you very much for putting forward this good comment! We added the description in the new revision (lines 275 to 277).

Reviewer 4 Report

Comments and Suggestions for Authors

Dear authors

the paper is quite interesting to promote inter-cropping tea with flowering cherry, in the production of the famous Lushan Yunwu green tea.

Main comments:

Line 23 (Abstract): please, add the meaning of the abbreviation HS-SPME/GC-MS

Line 95: A panel composed of five professional reviewers was employed. It would be better to have al least seven professional reviewers. 

2.2. Sensory Assessment: Could the authors improve the information on this subject?

In particular, can they add the description of the attributes employed by the panel? For example, "flowery aroma obvious", "Fresh and clear", and "Fresh, heavy and thick" for the taste, etc.

Clarify also the term employed in the "Comprehensive assessment".

Line174-175: please, improve this sentence, it is not clear.

Comments on the Quality of English Language

Dear authors,

please revise the use of the tenses of the English verbs in all the text.

Line 40: Recent studies have found better found

Line 49, line 51: has been found  better was found

Line 53, line 55: have shown better showed

Line 69: has received better received

Line 273: there is a comma after bloom, but it is not necessary. 

Line 293: have been shown to promote ... better showed

Line 294:  has shown that  better showed that

Line 305: ..have found,, better found

Line 309: Studies have shown better Sudies showed 

Author Response

Dear Reviewer,

Thanks a lot for your good suggestion. We find that the comments are of great help for the improvement of our manuscript. All of the comments have been considered in the revised manuscript, and all of the revisions are marked in yellow in the revised version. Below we have provided in detail the point-by-point replies to the comments.

Best wishes,

Yours sincerely,

Yanan Liu

Point 1: Line 23 (Abstract): please, add the meaning of the abbreviation HS-SPME/GC-MS.

Response: Thank you for your suggestion. We added the meaning of the abbreviation HS-SPME/GC-MS at line 23.

Point 2: Line 95: A panel composed of five professional reviewers was employed. It would be better to have at least seven professional reviewers.

2.2. Sensory Assessment: Could the authors improve the information on this subject?

In particular, can they add the description of the attributes employed by the panel? For example, "flowery aroma obvious", "Fresh and clear", and "Fresh, heavy and thick" for the taste, etc.

Clarify also the term employed in the "Comprehensive assessment".

Response: Thanks a lot for your good suggestion! We re-conducted the sensory evaluation between EG and CG tea samples, and the panel consisted of 9 people. Radar plot for comparative descriptive aroma profiles of EG and CG were added in the new revision (lines 97 to 114, lines 180 to 187).

Point 3: Line174-175: please, improve this sentence, it is not clear.

Response: Thank you very much for putting forward this good comment! I’m sorry for the confused expression. We have added the description in the new revision (lines 191 to 192).

Point 4: Please revise the use of the tenses of the English verbs in all the text.

Response: Thanks a lot! We revised the use of the tenses of the English verbs in all the text.

Round 2

Reviewer 1 Report

Comments and Suggestions for Authors

No additional comments to the first review.